

SciPost Phys. Lect. Notes 6 (2018)

# Nested Algebraic Bethe Ansatz in integrable models: recent results

**Stanislav Pakuliak [1] Eric Ragoucy [2] and Nikita Slavnov [3]**

**1** Laboratory of Theoretical Physics, JINR, Dubna, Moscow reg., Russia
**2** Laboratoire de Physique Théorique LAPTh, CNRS and USMB,
BP 110, 74941 Annecy-le-Vieux Cedex, France
**3** Steklov Mathematical Institute of Russian Academy of Sciences, Moscow, Russia

*This paper corresponds to two talks given by E.R. and N.S. at*
*"Correlation functions of quantum integrable systems and beyond,"*
*in honor of Jean-Michel Maillet for his 60s (ENS Lyon, October 2017)*

## Abstract

**We review the recent results we have obtained in the framework of algebraic Bethe ansatz based on algebras and superalgebras of rank greater than 1 or on their quantum deformation. We present different expressions (explicit, recursive or using the current realization of the algebra) for the Bethe vectors. Then, we provide a general expression (as sum over partitions) for their scalar products. For some particular cases (in the case of $gl(3)$ or its quantum deformation, or of $gl(2|1)$), we provide determinant expressions for the scalar products. We also compute the form factors of the monodromy matrix entries, and give some general methods to relate them. A coproduct formula for Bethe vectors allows to get the form factors of composite models.**

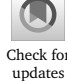
# 1 Introduction

Calculation of correlation functions is one of the most challenging problems in the study of quantum integrable models. Among various methods to solving this problem, we would like to mention the form factor approach in the framework of the algebraic Bethe ansatz (ABA). This approach was found to be very effective, in particular, in studying asymptotic behavior of correlation functions of critical models in the works by J.-M. Maillet and coauthors [1–5].

In the works listed above, the correlation functions of the Lieb–Liniger model and XXZ spin-1/2 chain were considered. From the point of view of the ABA, these are models respectively described by the Yangian $Y(gl_2)$ and the quantum group $U_q(\widehat{gl}_2)$. At the same time, there exist a lot of models of physical interest that are described by the algebras with higher rank of symmetries (see e.g. [6–9]). Our review is devoted to the recent results obtained in this field.

The first problem that one encounters when studying models with a high rank of symmetry is to construct the eigenvectors of the Hamiltonian. In the case under consideration, they have a much more complex form, in comparison with $gl_2$ based models [10–15]. First we need to build the so-called off-shell Bethe vectors (BVs) that depend on sets of complex parameters. If these parameters satisfy special constraints (Bethe ansatz equations), then the corresponding vector becomes an eigenvector of the quantum Hamiltonian (on-shell Bethe vector).

The second problem is the calculation of the scalar products of off-shell BVs. In the study of correlation functions, one can not confine himself to treating only on-shell Bethe vectors, since the actions of operators on states, generally speaking, transform on-shell BVs to linear combinations of off-shell BVs. In view of rather complex structure of BVs, their scalar products also were found to be difficult to compute [16, 17].

The third problem consists in calculating the form factors of local operators. Formally, it reduces to scalar products of BVs. The problem, however, is to obtain such representations for form factors that would be convenient for calculation of correlation functions. In particular, such representations include determinant formulas for form factors.

Finally, having convenient formulas for form factors, one can proceed to a direct calculation of the correlation functions with the framework of the form factorial expansion. It should be noted, however, that the procedure for summing the form factors strongly depends on the specific model. In other words, this procedure depends on the concrete representation of the algebra describing the given quantum model. At the same time, the first three problems can be formulated already at the level of the algebra, what makes it possible to obtain their solutions for a wide class of models within the framework of the ABA. Therefore, in this review, we will focus on the first three problems.

The plan of this presentation reflects the steps described above. We first present in section 2 the framework we work with, namely the generalized integrable models. Then, we show in section 3 some results on the construction of BVs for these models. Their scalar products will

be dealt in section 4, and they can be gathered in three main categories: a generalized sum formula, some determinant forms and a Gaudin determinant for the norm of BVs. To compute form factors (FF), we will use four methods: the twisted scalar product tricks, the zero mode method, the universal FF, and finally the composite model. They are presented in section 5. Finally, we will conclude on open problems. Since the calculations are rather technical we will focus on ideas and results, referring to the original papers for the details.

To ease the presentation we will mainly focus on the case of Yangians $Y(gl_{\mathfrak{n}})$, possibly fixing $\mathfrak{n} = 3$. However, after each result, we will precise to which extend these results can be used, and refer to the relevant publications where they can be found.

## 2 Generalized quantum integrable models

The construction of generalized quantum integrable models relies on two main ingredients: the $R$-matrix and the monodromy matrix.

**The $R$-matrix.** It depends on two spectral parameters $z_1, z_2 \in \mathbb{C}$ and acts on a tensor space $\mathbb{C}^{\mathfrak{n}} \otimes \mathbb{C}^{\mathfrak{n}}$, i.e. $R(z_1, z_2) \in V \otimes V$ with $V = End(\mathbb{C}^{\mathfrak{n}})$. $R(z_1, z_2)$ obeys the Yang–Baxter equation (YBE), written in $V \otimes V \otimes V$:

$$R^{12}(z_1, z_2) R^{13}(z_1, z_3) R^{23}(z_2, z_3) = R^{23}(z_2, z_3) R^{13}(z_1, z_3) R^{12}(z_1, z_2). \tag{1}$$

Here and below, we use the auxiliary space notation, where the exponent indicates on which copies of $\mathbb{C}^{\mathfrak{n}}$ acts the $R$-matrix, e.g. $R^{12} = R \otimes \mathbb{I}_{\mathfrak{n}}$, $R^{23} = \mathbb{I}_{\mathfrak{n}} \otimes R$, ...

**The universal monodromy matrix $T(z) \in V \otimes \mathscr{A}$.** It contains the generators of the algebra we will work with. In the present paper we will focus on (super)algebras $\mathscr{A} = Y(gl_{\mathfrak{n}})$, $U_q(\widehat{gl}_{\mathfrak{n}})$, $Y(gl_{\mathfrak{m}|\mathfrak{p}})$ and $U_q(\widehat{gl}_{\mathfrak{m}|\mathfrak{p}})$. Note however that the construction can be also done for other algebras, such as $\mathscr{A} = Y(so_{\mathfrak{n}})$, $Y(sp_{\mathfrak{n}})$, $U_q(\widehat{so}_{\mathfrak{n}})$, $U_q(\widehat{sp}_{\mathfrak{n}})$, $Y(osp_{\mathfrak{m}|\mathfrak{p}})$, $U_q(\widehat{osp}_{\mathfrak{m}|\mathfrak{p}})$. The algebraic structure of $\mathscr{A}$ is contained in the two following relations:

$$T(z) = \sum_{i,j=1}^{\mathfrak{n}} e_{ij} \otimes T_{ij}(z) \in V \otimes \mathscr{A}[[z^{-1}]], \tag{2}$$

$$R^{12}(z_1, z_2) T^1(z_1) T^2(z_2) = T^2(z_2) T^1(z_1) R^{12}(z_1, z_2). \tag{3}$$

The first relation shows how the generators[1] $T_{ij}(z)$ are encoded in a matrix. The second one (called RTT relation) provides the commutation relations of the algebra. Again, we have used the auxiliary space notation, i.e. $T^1(z_1) = T(z_1) \otimes \mathbb{I}_{\mathfrak{n}} \in V \otimes V \otimes \mathscr{A}$; $T^2(z_2) = \mathbb{I}_{\mathfrak{n}} \otimes T(z_2) \in V \otimes V \otimes \mathscr{A}$. We will note $\mathfrak{n} = \operatorname{rank}\mathscr{A}$ (i.e. $\mathfrak{n} = \mathfrak{m}$ or $\mathfrak{m} + \mathfrak{p}$, depending on the algebra we consider).

Remark that $T(z)$ is a *universal* monodromy matrix, meaning that the generators of $\mathscr{A}$ are not represented. What is usually called a monodromy matrix corresponds to $\pi(T(z))$ where $\pi$ is a representation morphism. The choice of a representation (hence of the morphism $\pi$) fixes the physical model we work with. In fact, most of the calculations can be done with mild assumptions on the type of representation used to define the model. This leads to the notion of generalized models, that are based on lowest weight representations without specifying the lowest weight.

---

[1]Strictly speaking, the generators of $\mathscr{A}$ are obtained through an expansion of $T_{ij}(z)$ in $z$, $z^{-1}$. However, the expansion depends on the algebra we are considering, and most of the calculations can be done using the generating functions $T_{ij}(z)$, so that we will loosely call them 'generators'.

**Choice of (lowest weight) representations of $\mathscr{A}$.** The generalized models are defined from the universal monodromy matrix, assuming that it obeys the additional relations:

$$T_{jj}(z)|0\rangle = \lambda_j(z)|0\rangle, \qquad j = 1,..,\mathfrak{n}, \qquad T_{ij}(z)|0\rangle = 0, \qquad 1 \leq j < i \leq \mathfrak{n}, \qquad (4)$$

where $|0\rangle$ is some reference state (the so-called pseudo-vacuum) and $\lambda_j(z)$, $j = 1,..,\mathfrak{n}$, are arbitrary functions. Up to normalisation of $T(z)$, we only need the ratios

$$r_j(z) = \frac{\lambda_j(z)}{\lambda_{j+1}(z)}, \qquad j = 1,...,\mathfrak{n}-1.$$

In generalized models, $r_j(z)$ are kept as free functional parameters. The calculations we present will be valid for arbitrary functions $r_j(z)$.

**The transfer matrix $\mathfrak{t}(z)$.** It encodes the dynamics of the model as well as its conserved quantities. For algebras, the transfer matrix is defined as

$$\mathfrak{t}(z) = \operatorname{tr} T(z) = T_{11}(z) + ... + T_{\mathfrak{n}\mathfrak{n}}(z). \qquad (5)$$

For superalgebras, it takes the form

$$\mathfrak{t}(z) \;=\; \operatorname{str} T(z) = \sum_{i=1}^{\mathfrak{n}} (-1)^{[i]} T_{ii}(z) \qquad (6)$$

$$\;=\; T_{11}(z) + ... + T_{\mathfrak{m}\mathfrak{m}}(z) - T_{\mathfrak{m}+1,\mathfrak{m}+1}(z) - ... - T_{\mathfrak{p}+\mathfrak{m},\mathfrak{p}+\mathfrak{m}}(z), \qquad (7)$$

where $[.]$ is the standard $\mathbb{Z}_2$ gradation used for superalgebras, implicitly defined in (7). Due to (3), we have $[\mathfrak{t}(z), \mathfrak{t}(z')] = 0$, so that the transfer matrix defines an integrable model (with periodic boundary conditions).

**Example: the "fundamental" spin chain.** To illustrate the different notions presented above, we consider the following monodromy matrix:

$$T^0(z|\bar{z}) \;=\; R^{01}(z-z_1)R^{02}(z-z_2)\cdots R^{0L}(z-z_L),$$

where $\bar{z} = \{z_1,...,z_L\}$ are complex parameters, called the inhomogeneities. $1,2,...,L$ are the quantum (physical) spaces of the spin chain, they are $\mathfrak{n}$-dimensional: on each site the "spins" can take $\mathfrak{n}$ values. The auxiliary space 0 has the same dimension.

Due to the YBE, one shows that the above monodromy matrix indeed obeys the RTT relation (3). The form of the $R$-matrix, for all the algebras $\mathscr{A} = Y(gl_\mathfrak{m})$, $U_q(\widehat{gl}_\mathfrak{m})$, $Y(gl_{\mathfrak{m}|\mathfrak{p}})$, $U_q(\widehat{gl}_{\mathfrak{m}|\mathfrak{p}})$ ensures that this monodromy matrix obeys the lowest property (4). For the Yangian $Y(gl_\mathfrak{n})$, the weights read

$$\lambda_1(z) \;=\; \prod_{\ell=1}^{L}\left(1 + \frac{c}{z - z_\ell}\right) \quad \text{and} \quad \lambda_j(z) = 1 \qquad j = 2,...,\mathfrak{n}.$$

For any algebra $\mathscr{A}$, it is the simplest spin chain that one can construct. It is built on the tensor product of $L$ fundamental representations of the underlying finite dimensional Lie algebra, and corresponds to a periodic spin chain with $L$ sites, each of them carrying a fundamental representation of $\mathscr{A}$.

**To illustrate the presentation, we will focus on the Yangian** $Y(gl_{\mathfrak{n}})$. Formulas will be displayed for this algebra, but we will mention when they exist for other algebras.

The Yangian $Y(gl_{\mathfrak{n}})$ has a rational $R$-matrix

$$
\begin{aligned}
R(z_1, z_2) &= \mathbf{I} + g(z_1, z_2)\mathbf{P} \in End(\mathbb{C}^{\mathfrak{n}}) \otimes End(\mathbb{C}^{\mathfrak{n}}), \\
g(z_1, z_2) &= \frac{c}{z_1 - z_2},
\end{aligned}
$$

where $\mathbf{I}$ is the identity matrix, $\mathbf{P}$ is the permutation matrix between two spaces $End(\mathbb{C}^{\mathfrak{n}})$, $c$ is a constant. It corresponds to XXX-like models and is based on $Y(gl_{\mathfrak{n}})$.

Explicitly, in the case $\mathfrak{n} = 3$, the $R$-matrix has the form

$$
R(z_1, z_2) = \left(\begin{array}{ccc|ccc|ccc}
f & 0 & 0 & 0 & 0 & 0 & 0 & 0 & 0 \\
0 & 1 & 0 & g_+ & 0 & 0 & 0 & 0 & 0 \\
0 & 0 & 1 & 0 & 0 & 0 & g_+ & 0 & 0 \\
\hline
0 & g_- & 0 & 1 & 0 & 0 & 0 & 0 & 0 \\
0 & 0 & 0 & 0 & f & 0 & 0 & 0 & 0 \\
0 & 0 & 0 & 0 & 0 & 1 & 0 & g_+ & 0 \\
\hline
0 & 0 & g_- & 0 & 0 & 0 & 1 & 0 & 0 \\
0 & 0 & 0 & 0 & 0 & g_- & 0 & 1 & 0 \\
0 & 0 & 0 & 0 & 0 & 0 & 0 & 0 & f
\end{array}\right),
$$

where $g_+ = g_- \equiv g(z_1, z_2)$ and $f \equiv f(z_1, z_2) = 1 + g(z_1, z_2)$. Note that the $R$-matrix for $U_q(\widehat{gl}_3)$ has a similar form, but with different functions $g_\pm$ and $f$.

## 2.1 Notation

We have already introduced the functions

$$
g(z_1, z_2) = \frac{c}{z_1 - z_2} \quad \text{and} \quad f(z_1, z_2) = \frac{z_1 - z_2 + c}{z_1 - z_2},
$$

that enter in the definition of the $R$-matrix, and describe the interaction in the bulk. The functions presented above are of XXX type. For completeness, we give below the functions $f$ and $g$ for the XXZ type:

$$
g(z_1, z_2) = \frac{q - q^{-1}}{z_1 - z_2} \quad \text{and} \quad f(z_1, z_2) = \frac{qz_1 - q^{-1}z_2}{z_1 - z_2}.
$$

We have also seen the free functionals

$$
r_i(z) = \frac{\lambda_i(z)}{\lambda_{i+1}(z)}, \qquad i = 1, \dots, \mathfrak{n} - 1,
$$

that (potentially) describe the representation used for the model. These are all the scalar functions we will deal with.

We will use many sets of variables and to lighten the presentation, we will use some notation for them:

- "bar" always denote sets of variables: $\bar{w}$, $\bar{u}$, $\bar{v}$ etc.

- Individual elements of the sets have latin subscripts: $w_j$, $u_k$, etc.

- # is the cardinality of a set: $\bar{w} = \{w_1, w_2\} \Rightarrow \#\bar{w} = 2$, etc.

- Subsets of variables are denoted by roman indices: $\bar{u}_{\mathrm{I}}$, $\bar{v}_{\mathrm{iv}}$, $\bar{w}_{\mathrm{II}}$, etc.

- Special case of subsets: $\bar{u}_j = \bar{u} \setminus \{u_j\}$, $\bar{w}_k = \bar{w} \setminus \{w_k\}$, etc.

Associated to these sets of variables, we use shorthand notation for products of scalar functions (when they depend on one or two variables). If a function depends on a set of variables, then one should take take the product over the sets, e.g.:

$$
\begin{aligned}
f(\bar{u}_{\mathrm{II}}, \bar{u}_{\mathrm{I}}) &= \prod_{u_j \in \bar{u}_{\mathrm{II}}} \prod_{u_k \in \bar{u}_{\mathrm{I}}} f(u_j, u_k), \\
r_1(\bar{u}_{\mathrm{II}}) &= \prod_{u_j \in \bar{u}_{\mathrm{II}}} r_1(u_j); \quad g(v_k, \bar{w}) = \prod_{w_j \in \bar{w}} g(v_k, w_j), \quad \text{etc.}
\end{aligned}
\tag{8}
$$

We use the same prescription for the products of commuting operators, for example,

$$
T_{jj}(\bar{u}_{\mathrm{I}}) = \prod_{u_j \in \bar{u}_{\mathrm{I}}} T_{jj}(u_j), \quad \text{etc.}
\tag{9}
$$

By definition, any product over the empty set is equal to 1. A double product is equal to 1 if at least one of the sets is empty.

## 3 Bethe vectors

### 3.1 Generalities

The framework to compute Bethe vectors has been developed by the Leningrad school in the 80's. It is the Nested Algebraic Bethe Ansatz (NABA), developed by Kulish and Reshetikhin [10, 11]. It provides vectors (the Bethe vectors, BVs) that depend on some parameters (the Bethe parameters) and that are eigenvectors of the transfer matrix provided the Bethe parameters obey some algebraic equations (the Bethe Ansatz Equations, BAEs). When it is the case, the BVs are called on-shell, while they are said off-shell otherwise. Our first goal is to provide explicit expressions for these BVs. The general strategy of the ABA is to start with the pseudo-vacuum vector $|0\rangle$, which is itself an eigenvector of the transfer matrix. Then, one applies the 'creation operators' $T_{ij}(u)$, $i < j$, on $|0\rangle$ to build more general vectors, and seek for combinations that can be transfer matrix eigenvectors.

**In the case of the "usual" XXX ($gl_2$) spin chain.** The construction of BVs is rather simple, since we have only one 'raising' operator $T_{12}(z)$:

$$
\mathbb{B}_a(\bar{u}) = T_{12}(u_1) T_{12}(u_2) \cdots T_{12}(u_a) |0\rangle,
\tag{10}
$$

which leads to one set of Bethe parameters $\bar{u} = \{u_1, ..., u_a\}$. Then, asking $\mathbb{B}_a(\bar{u})$ to be an eigenvector of the transfer matrix $\mathfrak{t}(z) = T_{11}(z) + T_{22}(z)$ leads to the BAE:

$$
r_1(u_j) = \frac{f(u_j, \bar{u}_j)}{f(\bar{u}_j, u_j)}, \quad j = 1, 2, ..., a.
$$

**In the case of higher rank $\mathfrak{n}$.** There are many raising operators $T_{ij}(z)$, $1 \le i < j \le \mathfrak{n}$, and the calculation becomes more tricky. In particular, there are $\mathfrak{n} - 1$ different sets of Bethe parameters:

$$
\begin{aligned}
\bar{t}^{(j)} &= \{t_1^{(j)}, ..., t_{a_j}^{(j)}\}, \quad \#\bar{t}^{(j)} = a_j \in \mathbb{Z}_{\ge 0}, \quad j = 1, 2, ..., \mathfrak{n} - 1, \\
\bar{t} &= \{\bar{t}^{(1)}, \bar{t}^{(2)}, ..., \bar{t}^{(\mathfrak{n}-1)}\}, \quad \bar{a} = \{a_1, a_2, ..., a_{\mathfrak{n}-1}\}.
\end{aligned}
$$

On needs to find how to put together all the raising operators, and $\mathbb{B}_{\bar{a}}(\bar{t})$ appears to be much more complicated. The expression of $\mathbb{B}_{\bar{a}}(\bar{t})$ is fixed by asking it to be a transfer matrix eigenvector

$$\mathfrak{t}(z)\mathbb{B}_{\bar{a}}(\bar{t}) = \tau(z|\bar{t})\mathbb{B}_{\bar{a}}(\bar{t}), \tag{11}$$

provided the Bethe equations are obeyed. For illustration, we give the eigenvalue and the BAEs in the case of the Yangian $Y(gl_{\mathfrak{n}})$ [10, 11]:

$$\tau(z|\bar{t}) = \sum_{i=1}^{\mathfrak{n}} \lambda_i(z) f(z, \bar{t}^{(i-1)}) f(\bar{t}^{(i)}, z), \tag{12}$$

$$r_i(\bar{t}_{\mathrm{I}}^{(i)}) = \frac{f(\bar{t}_{\mathrm{I}}^{(i)}, \bar{t}_{\mathrm{II}}^{(i)})}{f(\bar{t}_{\mathrm{II}}^{(i)}, \bar{t}_{\mathrm{I}}^{(i)})} \frac{f(\bar{t}^{(i+1)}, \bar{t}_{\mathrm{I}}^{(i)})}{f(\bar{t}_{\mathrm{I}}^{(i)}, \bar{t}^{(i-1)})}, \quad i = 1, ..., \mathfrak{n}-1, \tag{13}$$

with the convention that $\bar{t}^{(0)} = \emptyset = \bar{t}^{(\mathfrak{n})}$. Recall that here we use the shorthand notation (8) for the products of the functions $r_i$ and $f$. In particular, any product over the empty set equals 1. BAEs hold for arbitrary partitions of the sets $\bar{t}^{(i)}$ into subsets $\{\bar{t}_{\mathrm{I}}^{(i)}, \bar{t}_{\mathrm{II}}^{(i)}\}$.

**Dual Bethe vectors $\mathbb{C}_{\bar{a}}(\bar{t})$.** As already mentioned, $\mathbb{B}_{\bar{a}}(\bar{t})$ is a transfer matrix eigenvector provided the Bethe equations are obeyed. In the same way, one can construct dual BVs that are left eigenvectors of the transfer matrix

$$\mathbb{C}_{\bar{a}}(\bar{t})\mathfrak{t}(z) = \tau(z|\bar{t})\mathbb{C}_{\bar{a}}(\bar{t}),$$

provided the (same) BAEs are obeyed, and where $\tau(z|\bar{t})$ is the same as in (11). In that case they will be called on-shell dual BVs, and off-shell dual BVs otherwise.

Below, we will mainly focus on the BVs $\mathbb{B}_{\bar{a}}(\bar{t})$, but formulas also exist for dual Bethe vectors. A simple way to get such formulas is to use an anti-morphism $\psi$. For the Yangian $Y(gl_{\mathfrak{n}})$ it takes the form $\psi(T_{ij}(u)) = T_{ji}(u)$ and allows to define the dual BV as

$$\mathbb{C}_{\bar{a}}(\bar{t}) = \psi(\mathbb{B}_{\bar{a}}(\bar{t})).$$

- *An example of the use of the morphism $\psi$ in the Yangian case can be found in [18]. Note that in the case of super-Yangians, $\psi$ relates BVs of $Y(gl_{\mathfrak{m}|\mathfrak{p}})$ to dual BVs of $Y(gl_{\mathfrak{p}|\mathfrak{m}})$, see [19, 20]. The same is true for its generalization to the $U_q(\widehat{gl}_{\mathfrak{m}})$ algebra, see e.g. [21].*

**Generalized models.** Usually, when dealing with e.g. spin chain models, the Bethe equations are seen as a 'quantization' of the Bethe parameters $\bar{t}$. Here, for generalized models, since the functions $r_i(z)$ are not fixed, BAEs are rather viewed as functional relations between the functions $r_i(z)$, $i = 1, ..., \mathfrak{n}-1$ and the Bethe parameters $t_j^{(i)}$.

## 3.2 Expressions for Bethe vectors

There are different presentations for the BVs, each of them being adapted for different purpose.

**Known formulas: the trace formula.** It is the first general expression for BVs of higher rank algebras. Again, as an illustration, we present it in the case of the Yangian $Y(gl_3)$. For a BV $\mathbb{B}_{a,b}(\bar{u}; \bar{v})$, where $a = \#\bar{u}$ and $b = \#\bar{v}$, one introduces $a+b$ auxiliary spaces $V = End(\mathbb{C}^3)$. Then, the Bethe vector can be written as

$$\mathbb{B}_{a,b}(\bar{u}; \bar{v}) = \left(\lambda_2(\bar{u})\lambda_2(\bar{v})f(\bar{v}, \bar{u})\right)^{-1} \underbrace{\operatorname*{tr}_{a+b}\left(\overbrace{\mathbb{T}(\bar{u}; \bar{v})\mathbb{R}(\bar{u}; \bar{v})}^{\in Y(gl_3)\otimes V^{\otimes(a+b)}} e_{21}^{\otimes a} \otimes e_{32}^{\otimes b}\right)|0\rangle}_{\in Y(gl_3)},$$

where $e_{ij}$ are the $3 \times 3$ elementary matrices (acting in $\mathbb{C}^3$) with 1 at position $(i, j)$ and 0 elsewhere. The trace $\mathrm{tr}_{a+b}$ is taken over the $a + b$ auxiliary spaces, and $\mathbb{T}(\bar{u}, \bar{v})$ (resp. $\mathbb{R}(\bar{u}, \bar{v})$) is a product of monodromy matrices (resp. $R$-matrices):

$$
\begin{aligned}
\mathbb{T}(\bar{u}, \bar{v}) &= T^1(u_1) \cdots T^a(u_a) \, T^{a+1}(v_1) \cdots T^{a+b}(v_b), \\
\mathbb{R}(\bar{u}, \bar{v}) &= \left( R^{a,a+1}(u_a, v_1) \cdots R^{a,a+b}(u_a, v_b) \right) \cdots \left( R^{1,a+1}(u_1, v_1) \cdots R^{1,a+b}(u_1, v_b) \right),
\end{aligned}
$$

where we have used the auxiliary space notation, i.e. the exponents indicate in which auxiliary space(s) the matrices act.

- *The trace formula was introduced by Tarasov and Varchenko for $Y(gl_{\mathrm{m}})$ and $U_q(\widehat{gl}_{\mathrm{m}})$ algebras [12]. It has been generalized to superalgebras $Y(gl_{\mathrm{m|p}})$ and $U_q(\widehat{gl}_{\mathrm{m|p}})$ in [22].*

**Recursion formulas.** They allow to build BVs with a 'big' number of Bethe parameters from BVs having a smaller number of them. Again, in the $gl_2$ case (10), these recursion formulas are rather trivial, $\mathbb{B}_{a+1}(\bar{u}) = T_{12}(u_k)\mathbb{B}_a(\bar{u}_k)$, while they become more intricate for higher ranks. In the case of $Y(gl_3)$, they take the form:

$$
\lambda_2(u_k) f(\bar{v}, u_k) \mathbb{B}_{a+1,b}(\bar{u}; \bar{v}) = T_{12}(u_k) \mathbb{B}_{a,b}(\bar{u}_k; \bar{v})
$$
$$
+ \sum_{i=1}^{b} r_2(v_i) g(v_i, u_k) f(\bar{v}_i, v_i) T_{13}(u_k) \mathbb{B}_{a,b-1}(\bar{u}_k; \bar{v}_i), \tag{14}
$$

$$
\lambda_3(v_k) f(v_k, \bar{u}) \mathbb{B}_{a,b+1}(\bar{u}; \bar{v}) = T_{23}(v_k) \mathbb{B}_{a,b}(\bar{u}; \bar{v}_k)
$$
$$
+ \sum_{j=1}^{a} g(v_k, u_j) f(u_j, \bar{u}_j) T_{13}(v_k) \mathbb{B}_{a-1,b}(\bar{u}_j; \bar{v}_k). \tag{15}
$$

Remark that considering the underlying finite Lie algebra $gl_3$ with simple roots $\alpha, \beta$, one sees that $\mathbb{B}_{a,b}(\bar{u}; \bar{v})$ "behaves" as the root $a\,\alpha + b\,\beta$. This reflects the fact that BVs are eigenvectors of the zero modes $T_{kk}[0]$, see section 5.2.

- *Recursion formulas are in fact a particular case of multiple action of $T_{ij}(\bar{x})$ on BVs. The case of $Y(gl_3)$ can be found in [23], $U_q(\widehat{gl}_3)$ in [24], and $Y(gl_{2|1})$ in [25]. It exists also for $Y(gl_{\mathrm{m|p}})$ [26] and $U_q(\widehat{gl}_{\mathrm{n}})$ [27].*

**Explicit formulas.** Solving the recursion relations, we obtain different explicit formulas for the Bethe vectors, which depend on the recursion we use, e.g. (14) or (15) in the $Y(gl_3)$ case. An example of such explicit expression is given by:

$$
\mathbb{B}_{a,b}(\bar{u}; \bar{v}) = \sum \frac{\lambda_2(\bar{v}_{\mathrm{I}}) \mathsf{K}_k(\bar{v}_{\mathrm{I}}|\bar{u}_{\mathrm{I}})}{\lambda_3(\bar{v})\lambda_2(\bar{u})} \frac{f(\bar{v}_{\mathrm{II}}, \bar{v}_{\mathrm{I}}) f(\bar{u}_{\mathrm{II}}, \bar{u}_{\mathrm{I}})}{f(\bar{v}_{\mathrm{II}}, \bar{u}) f(\bar{v}_{\mathrm{I}}, \bar{u}_{\mathrm{I}})} T_{12}(\bar{u}_{\mathrm{I}}) T_{13}(\bar{u}_{\mathrm{I}}) T_{23}(\bar{v}_{\mathrm{II}}) |0\rangle, \tag{16}
$$

where the sums are taken over partitions of the sets: $\bar{u} \Rightarrow \{\bar{u}_{\mathrm{I}}, \bar{u}_{\mathrm{II}}\}$ and $\bar{v} \Rightarrow \{\bar{v}_{\mathrm{I}}, \bar{v}_{\mathrm{II}}\}$ with $0 \le \#\bar{u}_{\mathrm{I}} = \#\bar{v}_{\mathrm{I}} = k \le \min(a, b)$ and $\mathsf{K}_k(\bar{v}_{\mathrm{I}}|\bar{u}_{\mathrm{I}})$ is the *Izergin–Korepin determinant*

$$
\mathsf{K}_k(\bar{x}|\bar{y}) = \Delta_k(\bar{x}) \Delta'_k(\bar{y}) \frac{f(\bar{x}, \bar{y})}{g(\bar{x}, \bar{y})} \det_k \left[ \frac{g^2(x_i, y_j)}{f(x_i, y_j)} \right], \tag{17}
$$

$$
\Delta_k(\bar{x}) = \prod_{\ell < m}^{k} g(x_\ell, x_m) \quad ; \quad \Delta'_k(\bar{y}) = \prod_{\ell < m}^{k} g(y_m, y_\ell). \tag{18}
$$

- *A fully explicit expression for BVs in the case of $Y(gl_3)$ was presented in [23] and in [27] for $U_q(\widehat{gl}_{\mathrm{n}})$. The generalization to superalgebras can be found in [28] for $Y(gl_{2|1})$ and in [26] for $Y(gl_{\mathrm{m|p}})$.*

**Current presentation and projection method.** Instead of presenting the algebra $\mathscr{A}$ in term of a monodromy matrix $T(z)$, one can use the current realization. It exists for the quantum groups $U_q(\widehat{gl}_n)$ and $U_q(gl_{m|p})$, as well as for the double Yangians $DY(gl_n)$ and $DY(gl_{m|p})$. The current realization is related to a Gauss decomposition of the monodromy matrix $T(z)$ [29]. Using the projection method introduced by Khoroshkin, Pakuliak, and collaborators in the years 2006-10, one gets an explicit expression of BVs in a different basis.

As an illustration of the projection method, we consider the current realization of $DY(gl_3)$. Then, BVs can be written as

$$\mathbb{B}_{a,b}(\bar{u};\bar{v}) = \mathscr{N}\,\mathscr{P}_f^+\Big(F_1(u_1)\cdots F_1(u_a)F_2(v_1)\cdots F_2(v_b)\Big)\,k_1(\bar{u})\,k_2(\bar{v})\,|0\rangle\,,$$

where

$$\mathscr{N} = \frac{\prod_{1\le j<i\le a} f(u_j,u_i)\prod_{1\le j<i\le b} f(v_j,v_i)}{\lambda_2(\bar{u})\lambda_3(\bar{v})f(\bar{v},\bar{u})}\,,$$

and

1. $k_1(z)$ and $k_2(z)$ are the Cartan generators;

2. $F_1(z)$ is the generator associated to the first simple (negative) root;

3. $F_2(z)$ is the generator associated to the second simple (negative) root;

4. $\mathscr{P}_f^+$ is the projector of the Borel subalgebra on the positive modes.

- *The construction of BV in the current presentation has been initiated for the $U_q(\widehat{gl}_n)$ algebra in [13–15] and then generalized to the (super)Yangian $Y(gl_{m|p})$ case in [26].*

### 3.3 Relations between the different expressions of Bethe vectors

All the formulas presented in section 3.2 are related:

1. The explicit expressions solve the recursion formulas;

2. The trace formula obeys the recursion formulas;

3. The recursion formulas uniquely fix the BVs, once the initial values

$$\mathbb{B}_{a,0}(\bar{u};\emptyset) = \frac{T_{12}(\bar{u})}{\lambda_2(\bar{u})}|0\rangle,\quad\text{or}\quad \mathbb{B}_{0,b}(\emptyset;\bar{v}) = \frac{T_{23}(\bar{v})}{\lambda_3(\bar{v})}|0\rangle$$

for $Y(gl_3)$ are known;

4. The projection of currents coincides with the trace formula.

Thus, they all describe the same (off-shell) BVs.

**Normalization of BVs.** The main property of BVs is that they become eigenvectors of the transfer matrix if the Bethe parameters enjoy the BAEs. Since any eigenvector is defined up to a normalization factor, the BVs also have a freedom in their normalization. The choice of normalization is a question of convenience. In the above formulas, the normalization was chosen as follows.

It follows from the explicit representation (16) that BV is a polynomial in $T_{ij}$ ($i < j$) acting on $|0\rangle$. Among the terms of this polynomial, there is one term that does not depend on the operator $T_{13}$. We call this monomial *main term* and denote by $\widetilde{\mathbb{B}}_{a,b}(\bar{u};\bar{v})$. Thus,

$$\mathbb{B}_{a,b}(\bar{u};\bar{v}) = \widetilde{\mathbb{B}}_{a,b}(\bar{u};\bar{v}) + \dots\,, \tag{19}$$

where ellipsis refers to all the terms containing at least one operator $T_{13}$, and

$$\widetilde{\mathbb{B}}_{a,b}(\bar{u};\bar{v}) = \frac{T_{12}(\bar{u})T_{23}(\bar{v})|0\rangle}{\lambda_3(\bar{v})\lambda_2(\bar{u})f(\bar{v},\bar{u})}. \tag{20}$$

Thus, we fix the normalization of BV by the explicit form of the main term. This normalization is convenient for recursion formulas, formulas of the action of $T_{ij}(z)$ on BVs, calculation of scalar products of BVs.

We use similar conventions on the normalization in the cases $Y(gl_\mathfrak{n})$, $U_q(\widehat{gl}_\mathfrak{n})$, and $Y(gl_{\mathfrak{m}|\mathfrak{p}})$. In all these cases the BV contain a term that depends on the operators $T_{i,i+1}$ only. We call it the main term. In the case of $Y(gl_\mathfrak{n})$ it is normalized as follows:

$$\widetilde{\mathbb{B}}_{\bar{a}}(\bar{t}) = \frac{T_{12}(\bar{t}^{(1)})T_{23}(\bar{t}^{(2)})\dots T_{\mathfrak{n}-1,\mathfrak{n}}(\bar{t}^{(\mathfrak{n}-1)})|0\rangle}{\prod_{i=1}^{\mathfrak{n}-1}\lambda_{i+1}(\bar{t}^{(i)})\prod_{i=1}^{\mathfrak{n}-2}f(\bar{t}^{(i+1)},\bar{t}^{(i)})}. \tag{21}$$

In the case $U_q(\widehat{gl}_\mathfrak{n})$ the normalization is the same, but one should take $q$-deformed analogs of the $f$-functions. For the superalgebra $Y(gl_{\mathfrak{m}|\mathfrak{p}})$, the normalization looks similar, but it takes into account the grading (see [20]).

## 4 Scalar products

Once the BVs (and dual BVs) are constructed, one can consider their scalar product

$$S_{\bar{a}}(\bar{s}|\bar{t}) = \mathbb{C}_{\bar{a}}(\bar{s})\mathbb{B}_{\bar{a}}(\bar{t}), \tag{22}$$

where

$$\begin{aligned}\bar{s} &= \{\bar{s}^{(1)},\bar{s}^{(2)},\dots\bar{s}^{(\mathfrak{n}-1)}\}, \\ \bar{t} &= \{\bar{t}^{(1)},\bar{t}^{(2)},\dots\bar{t}^{(\mathfrak{n}-1)}\}\end{aligned} \qquad \#\bar{s}^{(j)} = \#\bar{t}^{(j)}, \quad j = 1,\dots,\mathfrak{n}-1. \tag{23}$$

If $\#\bar{s}^{(j)} \neq \#\bar{t}^{(j)}$ for at least one $j$, then the scalar product vanishes.

### 4.1 Sum formula

The scalar product of generic off-shell BVs can be presented in the form known as a *sum formula*

$$S_{\bar{a}}(\bar{s}|\bar{t}) = \sum W_{\mathrm{part}}(\bar{s}_{\mathrm{I}},\bar{s}_{\mathrm{II}}|\bar{t}_{\mathrm{I}},\bar{t}_{\mathrm{II}})\prod_{j=1}^{\mathfrak{n}-1} r_j(\bar{s}_{\mathrm{I}}^{(j)})r_j(\bar{t}_{\mathrm{II}}^{(j)}). \tag{24}$$

Korepin and then Reshetikhin were the first to obtain such formula, see references below. In (24), the sum is taken over all possible partitions of each set $\bar{t}^{(j)}$ and $\bar{s}^{(j)}$ into subsets $\{\bar{t}_{\mathrm{I}}^{(j)},\bar{t}_{\mathrm{II}}^{(j)}\}$ and $\{\bar{s}_{\mathrm{I}}^{(j)},\bar{s}_{\mathrm{II}}^{(j)}\}$ respectively, such that $\#\bar{t}_{\mathrm{I}}^{(j)} = \#\bar{s}_{\mathrm{I}}^{(j)}$. The dependence on the monodromy matrix vacuum eigenvalues $r_j$ is given explicitly. The coefficient $W_{\mathrm{part}}$ are rational functions of the Bethe parameters $\bar{s}$ and $\bar{t}$. They are completely determined by the $R$-matrix. Thus, they do not depend on the specific representative of the generalized model.

The first formula of this type, corresponding to the $Y(gl_2)$ and $U_q(\widehat{gl}_2)$ based models, was obtained by Korepin. For these models, one can derive the sum formula using the explicit form of the BVs. However, in the models with higher rank of symmetry, the use of explicit formulas for BVs leads to too cumbersome expressions. A generalization of the sum formula to the $Y(gl_3)$ case was done by Reshetikhin via a special diagram technique. In the case of $Y(gl_\mathfrak{n})$, $Y(gl_{\mathfrak{m}|\mathfrak{p}})$, and $U_q(\widehat{gl}_\mathfrak{m})$ the sum formula was derived by the use of a coproduct formula

for BVs [20]. This method allows to express an arbitrary coefficient $W_{\text{part}}$ in terms of so-called highest coefficients. Namely, if we set

$$
\begin{aligned}
Z(\bar{s}|\bar{t}) &= W_{\text{part}}(\bar{s}, \emptyset | \bar{t}, \emptyset)\,, \\
\hat{Z}(\bar{s}|\bar{t}) &= W_{\text{part}}(\emptyset, \bar{s} | \emptyset, \bar{t})\,,
\end{aligned}
\tag{25}
$$

then the general coefficient $W_{\text{part}}(\bar{s}_{\text{I}}, \bar{s}_{\text{II}} | \bar{t}_{\text{I}}, \bar{t}_{\text{II}})$ has the following form

$$
W_{\text{part}}(\bar{s}_{\text{I}}, \bar{s}_{\text{II}} | \bar{t}_{\text{I}}, \bar{t}_{\text{II}}) = Z(\bar{s}_{\text{I}}|\bar{t}_{\text{I}})\, \hat{Z}(\bar{s}_{II}|\bar{t}_{II}) \, \frac{\prod_{k=1}^{\text{n}-1} f(\bar{s}_{\text{II}}^{(k)}, \bar{s}_{\text{I}}^{(k)}) f(\bar{t}_{\text{I}}^{(k)}, \bar{t}_{\text{II}}^{(k)})}{\prod_{j=1}^{\text{n}-2} f(\bar{s}_{\text{II}}^{(j+1)}, \bar{s}_{\text{I}}^{(j)}) f(\bar{t}_{\text{I}}^{(j+1)}, \bar{t}_{\text{II}}^{(j)})}.
\tag{26}
$$

The highest coefficients are known explicitly for $Y(gl_3)$, $Y(gl_{2|1})$, and $U_q(\widehat{gl}_3)$. For higher rank algebras they can be constructed via special recursions.

- *We already mentioned Korepin [30] for $Y(gl_2)$ or $U_q(\widehat{gl}_2)$, and Reshetikhin [16] for $Y(gl_3)$. For $U_q(\widehat{gl}_3)$, the highest coefficient is given in [31], and the full formula in [32]. The super Yangian $Y(gl_{2|1})$ was dealt in [33], while the general cases of $Y(gl_{\text{m}|\mathfrak{p}})$ and $U_q(\widehat{gl}_\text{n})$ were respectively presented in [20] and [21].*

The expression (24) is valid for all BVs (on-shell or off-shell). However, it is difficult to handle, specially when considering the thermodynamic limit, so that we look for determinant expressions for $S_{\bar{a}}(\bar{s}|\bar{t})$.

## 4.2 Determinant formula

It is known that for the $Y(gl_2)$ and $U_q(\widehat{gl}_2)$ based models the sum over partitions in (24) can be reduced to a single determinant if one of BVs is on-shell [34]. An analog of this determinant representation for the higher rank algebras is not known for today. However, determinant formulas for the scalar products have been obtained in some particular cases. One needs to impose more restrictive conditions for the BVs as we shall see below. The results have been obtained only for some specific algebras that we describe at the end of this subsection.

Consider the particular case $Y(gl_3)$ and the scalar product of an on-shell Bethe vector $\mathbb{B}_{a,b}(\bar{u}^B; \bar{v}^B)$ with a *twisted* dual on-shell Bethe vector $\mathbb{C}_{a,b}^{\kappa}(\bar{u}^C; \bar{v}^C)$. To define the twisted dual on-shell Bethe vector we consider the twisted transfer matrix

$$
\mathfrak{t}_\kappa(z) = \text{tr}\big(M\, T(z)\big) = T_{11}(z) + \kappa T_{22}(z) + T_{33}(z) \quad \text{with} \quad M = \text{diag}\{1, \kappa, 1\}.
$$

The twisted dual BV is an eigenvector of $\mathfrak{t}_\kappa(z)$

$$
\mathbb{C}_{a,b}^{\kappa}(\bar{u}^C; \bar{v}^C)\, \mathfrak{t}_\kappa(z) = \tau_\kappa(z|\bar{u}^C, \bar{v}^C)\, \mathbb{C}_{a,b}^{\kappa}(\bar{u}^C; \bar{v}^C),
\tag{27}
$$

with

$$
\tau_\kappa(z|\bar{u}^C, \bar{v}^C) = \lambda_1(z) f(\bar{u}^C, z) + \kappa \lambda_2(z) f(z, \bar{u}^C) f(\bar{v}^C, z) + \lambda_3(z) f(z, \bar{v}^C),
\tag{28}
$$

provided the twisted BAEs are satisfied:

$$
\begin{aligned}
r_1(u_j^c) &= \kappa \frac{f(u_j^c, \bar{u}_j^c)}{f(\bar{u}_j^c, u_j^c)} f(\bar{v}, u_j^c), \\
r_2(v_j^c) &= \frac{1}{\kappa} \frac{f(v_j^c, \bar{v}_j^c)}{f(\bar{v}_j^c, v_j^c)} \frac{1}{f(v_j^c, \bar{u})}.
\end{aligned}
$$

Then, the scalar product

$$S_{a,b}^{\kappa} \equiv \mathbb{C}_{a,b}^{\kappa}(\bar{u}^C; \bar{v}^C)\mathbb{B}_{a,b}(\bar{u}^B; \bar{v}^B)$$

can be written as:

$$S_{a,b}^{\kappa} = \frac{g^2(\bar{v}^C, \bar{u}^B)\Delta_a'(\bar{u}^C)\Delta_a(\bar{u}^B)\Delta_b'(\bar{v}^C)\Delta_b(\bar{v}^B)}{\kappa^b f(\bar{v}^C, \bar{u}^B)f(\bar{v}^C, \bar{u}^C)f(\bar{v}^B, \bar{u}^B)} \det_{a+b}\mathcal{M}, \tag{29}$$

where $\Delta_n$ and $\Delta_n'$ are given by (18), and $\mathcal{M}$ is a $(a+b)\times(a+b)$ matrix. If we set $\bar{\xi} = \{\bar{u}^B, \bar{v}^C\}$, then

$$\begin{aligned}
\mathcal{M}_{j,k} &= \frac{c}{\lambda_2(\xi_k)g(\xi_k, \bar{u}^C)g(\bar{v}^C, \xi_k)}\frac{\partial \tau_\kappa(\xi_k|\bar{u}^C, \bar{v}^C)}{\partial u_j^C}, \quad j = 1, \ldots, a, \\
\mathcal{M}_{a+j,k} &= \frac{-c}{\lambda_2(\xi_k)g(\xi_k, \bar{u}^B)g(\bar{v}^B, \xi_k)}\frac{\partial \tau(\xi_k|\bar{u}^B, \bar{v}^B)}{\partial v_j^B}, \quad j = 1, \ldots, b.
\end{aligned} \tag{30}$$

It is worth mentioning that in spite of this determinant representation is valid only for very specific case of the scalar product, it can be used as a generating formula for determinant representations of all form factors of the monodromy matrix entries (see sections 5.1, 5.2).

- These formulas can be found in [35] for the Yangian $Y(gl_3)$ (see also [36] for a determinant form of the highest coefficient). Similar determinant formula exists for the models described by $U_q(\widehat{gl}_3)$ algebra [37]. In the case of super-Yangians $Y(gl_{2|1})$ and $Y(gl_{1|2})$, a determinant representation was found for arbitrary diagonal twist matrix $M = \text{diag}\{\kappa_1, \kappa_2, \kappa_3\}$ [38]. In the case of the Yangian $Y(gl_3)$ and a twist matrix $M = \text{diag}\{\kappa_1, \kappa_2, \kappa_3\}$, a determinant formula for the scalar product was found up to corrections in $(\kappa_i - 1)(\kappa_j - 1)$ [39].

Unfortunately, for models with higher rank symmetry determinant representations are not known, except for the norms of on-shell BVs that we present now.

## 4.3 Norm of on-shell BVs: Gaudin determinant

In this section we give a determinant formula for the norm of an on-shell BV. The case of the models described by $Y(gl_2)$ and $U_q(\widehat{gl}_2)$ algebras was considered in [30], where a Gaudin hypothesis (see [40], [41]) was proved. A generalization of this result to the $Y(gl_3)$ based models was given in [16]. Here we focus on the $Y(gl_n)$ case to lighten the presentation.

**The Gaudin matrix.** To introduce the Gaudin matrix, we first rewrite the BAEs as $\Phi_k^{(i)} = 1$, $k = 1, \ldots, a_i$, $i = 1, \ldots, \mathfrak{n}-1$, where

$$\Phi_k^{(i)} = r_i(t_k^{(i)})\frac{f(\bar{t}_k^{(i)}, t_k^{(i)})}{f(t_k^{(i)}, \bar{t}_k^{(i)})}\frac{f(t_k^{(i)}, \bar{t}^{(i-1)})}{f(\bar{t}^{(i+1)}, t_k^{(i)})}, \qquad \begin{array}{l} k = 1, \ldots, a_i \\ i = 1, \ldots, \mathfrak{n}-1. \end{array} \tag{31}$$

Then, the Gaudin matrix $G$ is a block matrix $\left(G^{(i,j)}\right)_{i,j=1,\ldots,\mathfrak{n}-1}$, where each block $G^{(i,j)}$, of size $a_i \times a_j$, has entries

$$G_{k,l}^{(i,j)} = -c \frac{\partial \log(\Phi_k^{(i)})}{\partial t_l^{(j)}}. \tag{32}$$

**Norm of $\mathbb{B}_{\bar{a}}(\bar{t})$.** For an on-shell $\mathbb{B}_{\bar{a}}(\bar{t})$, the square of its norm is traditionally defined as $S_{\bar{a}}(\bar{t}) = \mathbb{C}_{\bar{a}}(\bar{t})\mathbb{B}_{\bar{a}}(\bar{t})$, where $\mathbb{C}_{\bar{a}}(\bar{t})$ is its dual BV. Then one has:

$$S_{\bar{a}}(\bar{t}) = \prod_{i=1}^{\mathfrak{n}}\prod_{k=1}^{a_i}\Big(\frac{f(\bar{t}_k^{(i)}, t_k^{(i)})}{f(\bar{t}^{(i+1)}, t_k^{(i)})}\Big) \det G\,, \tag{33}$$

where $\mathbb{B}_{\bar{a}}(t)$ is normalized as in (19) and (21). Note that if $\mathbb{B}_{\bar{a}}(t)$ and $\mathbb{C}_{\bar{a}}(s)$ are on-shell, we have $\mathbb{C}_{\bar{a}}(\bar{s})\mathbb{B}_{\bar{a}}(\bar{t}) = \delta_{\bar{s},\bar{t}}\, S_{\bar{a}}(\bar{t})$.

- *The representation of the norm of BVs are described in [42] for $Y(gl_{\mathfrak{n}})$ and $Y(gl_{\mathfrak{m}|\mathfrak{p}})$. Similar representations for $U_q(\widehat{gl}_{\mathfrak{n}})$ can be found in [21].*

# 5 Form factors (FF)

Form factors are the building blocks to study correlation functions. Here we will consider the FF of the monodromy matrix entries:

$$\mathscr{F}_{ij}(z|\bar{s};\bar{t}) = \mathbb{C}_{\bar{a}'}(\bar{s})T_{ij}(z)\mathbb{B}_{\bar{a}}(\bar{t}), \qquad i,j = 1,...,\mathfrak{n}-1$$

where both $\mathbb{C}_{\bar{a}'}(\bar{s})$ and $\mathbb{B}_{\bar{a}}(\bar{t})$ are on-shell BVs. The cardinalities of the Bethe parameters of the dual BV $\bar{a}' = \{a'_1,...,a'_{\mathfrak{n}}\}$ depend on the operator $T_{ij}(z)$. Since the FF is based on the monodromy matrix, we will call diagonal (resp. off-diagonal) the FF related to diagonal (resp. off-diagonal) entries of $T(z)$. To compute these FF, we use four different techniques:

1. The twisted scalar product trick (which leads to diagonal FF);

2. The zero mode method (to deduce off-diagonal FF);

3. The universal FF (for the general form of the FF);

4. The composite model (for FF of local operators).

We describe all these techniques below, again in the case of the Yangian $Y(gl_{\mathfrak{n}})$ to give simple formulas.

## 5.1 Twisted scalar product trick

Diagonal FF $\mathscr{F}_{jj}(z|\bar{s};\bar{t})$ are computed using the "twisted scalar product" trick. Consider a twist matrix $M = \text{diag}\{\kappa_1,...,\kappa_{\mathfrak{n}}\}$ and define a twisted transfer matrix as

$$\mathfrak{t}_{\bar{\kappa}}(z) = \text{tr}\big(MT(z)\big). \tag{34}$$

From the simple identity

$$\begin{aligned} \mathfrak{t}_{\bar{\kappa}}(z) - \mathfrak{t}(z) &= (\kappa_1 - 1)T_{11}(z) + \cdots + (\kappa_{\mathfrak{n}} - 1)T_{\mathfrak{n}\mathfrak{n}}(z), \\ T_{jj}(z) &= \frac{d}{d\kappa_j}\big(\mathfrak{t}_{\bar{\kappa}}(z) - \mathfrak{t}(z)\big), \quad j = 1,2,...,\mathfrak{n}, \end{aligned}$$

one deduces that

$$\mathscr{F}_{jj}(z|\bar{s};\bar{t}) = \frac{d}{d\kappa_j}\Big[\mathbb{C}_{\bar{a}}^{\bar{\kappa}}(\bar{s})\big(\mathfrak{t}_{\bar{\kappa}}(z) - \mathfrak{t}(z)\big)\mathbb{B}_{\bar{a}}(\bar{t})\Big]_{\bar{\kappa}=1} = \frac{d}{d\kappa_j}\Big[\big(\tau_{\bar{\kappa}}(z;\bar{s}) - \tau(z;\bar{t})\big)S_{\bar{a}}^{\bar{\kappa}}(\bar{s}|\bar{t})\Big]_{\bar{\kappa}=1}, \tag{35}$$

where $\bar{\kappa} = 1$ means that $\kappa_j = 1$ for $j = 1,...,\mathfrak{n}$. The function $\tau_{\bar{\kappa}}(z;\bar{s})$ is the eigenvalue of the twisted dual on-shell BV $\mathbb{C}_{\bar{a}}^{\bar{\kappa}}(\bar{s})$. It is given by equation (12), in which one should replace $\bar{t}^{(j)} \to \bar{s}^{(j)}$ and $\lambda_j(z) \to \kappa_j\lambda_j(z)$. Hence, if one knows the form of the twisted scalar product, one can deduce the diagonal FF.

- *The same trick also can be done for the cases $Y(gl_{m|p})$ and $U_q(\widehat{gl}_n)$. However, determinant representations for the twisted scalar products $S_{\bar{a}}^{\bar{\kappa}}(\bar{s}|\bar{t})$ for today are rather seldom, see section 4.2. They provide determinant expressions for diagonal FF in $Y(gl_3)$ and $Y(gl_{2|1})$ models, see [39] and [19] respectively. For $U_q(\widehat{gl}_3)$ models, due to the special twist, only $\mathcal{F}_{2,2}(z|\bar{s};\bar{t})$ is known [37]. Other FF are yet missing, but progress continues, and we hope to produce soon new formulas for diagonal FF.*

## 5.2 Zero mode method

**Zero modes of the monodromy matrix.** They correspond to the finite dimensional Lie subalgebra embedded in $\mathscr{A}$. For instance, they form a $gl_n$ Lie subalgebra in $Y(gl_n)$. Typically they are defined as

$$T_{ij}[0] = \lim_{w \to \infty} \frac{w}{c}\left(T_{ij}(w) - \delta_{ij}\right), \tag{36}$$

but depending on the model and on $\mathscr{A}$, some normalisation can be implied before taking the limit $w \to \infty$. The monodromy matrix is a representation of this Lie subalgebra:

$$\begin{aligned}
\left[T_{ij}[0], T_{kl}[0]\right] &= \delta_{kj} T_{il}[0] - \delta_{il} T_{kj}[0], \\
\left[T_{ij}[0], T_{kl}(z)\right] &= \delta_{kj} T_{il}(z) - \delta_{il} T_{kj}(z).
\end{aligned} \tag{37}$$

**Bethe vectors and zero modes.** The zero modes occur naturally in the BVs when one of the Bethe parameter is sent to infinity:

$$\begin{aligned}
\lim_{w \to \infty} \frac{w}{c} \mathbb{B}(\bar{t}^{(1)}, .., \{\bar{t}^{(j-1)}, w\}, \bar{t}^{(j)}, ..\bar{t}^{(n-1)}) &= T_{j-1,j}[0]\,\mathbb{B}(\bar{t}), \\
\lim_{w \to \infty} w\, \mathbb{C}(\bar{s}^{(1)}, .., \{\bar{s}^{(j-1)}, w\}, \bar{s}^{(j)}, ..\bar{s}^{(n-1)}) &= \mathbb{C}(\bar{s})\, T_{j,j-1}[0].
\end{aligned} \tag{38}$$

Here and further, to simplify the formulas. we omit the subscripts of the BVs that refer to the cardinalities of the Bethe parameters.

In the $Y(gl_n)$ and $Y(gl_{m|p})$ cases, the BAEs are compatible with the limit[2] $t_k^{(j-1)} \to \infty$ for $j$ and $k$ fixed. This implies that if the BV $\mathbb{B}(\bar{t})$ is on-shell then so is $\mathbb{B}(\{\infty, \bar{t}\})$.

Moreover, still for the $Y(gl_n)$ and $Y(gl_{m|p})$ cases, on-shell BVs obey a highest weight property with respect to the zero modes. Indeed, if $\mathbb{B}(\bar{t})$ and $\mathbb{C}(\bar{s})$ are on-shell, with $\bar{t}$ and $\bar{s}$ finite, then

$$T_{ij}[0]\mathbb{B}(\bar{t}) = 0 \quad \text{and} \quad \mathbb{C}(\bar{s})\,T_{ji}[0] = 0, \qquad i > j.$$

From these properties, we can elaborate a method to relate different FF. For obvious reason, we call it the zero mode method.

**The zero mode method ($Y(gl_n)$ and $Y(gl_{m|p})$ cases).** The basic idea behind the zero mode method is to use the Lie algebra symmetry generated by the zero modes and the highest weight property of on-shell BVs to obtain relations among form factors. To illustrate the method we

---

[2]To provide the compatibility of BAEs in this limit, one should have $r_j(z) \to 1$ as $z \to \infty$. This is not always true even for the models described by the $Y(gl_n)$ and $Y(gl_{m|p})$. We show in section 5.3 how this problem can be solved.

show it on an example in the $Y(gl_n)$ case, starting from a diagonal FF. We have

$$\lim_{w\to\infty} \frac{w}{c} \mathcal{F}_{jj}(z|\bar{s}; \bar{t}^{(1)}, .., \{\bar{t}^{(j-1)}, w\}, \bar{t}^{(j)}, ..\bar{t}^{(n-1)})$$

$$= \mathbb{C}(\bar{s})\, T_{jj}(z) \lim_{w\to\infty} \frac{w}{c} \mathbb{B}(\bar{t}^{(1)}, .., \{\bar{t}^{(j-1)}, w\}, \bar{t}^{(j)}, ..\bar{t}^{(n-1)})$$

$$= \mathbb{C}(\bar{s})\, T_{jj}(z)\, T_{j-1,j}[0]\, \mathbb{B}(\bar{t})$$

$$= \mathbb{C}(\bar{s}) \big[ T_{jj}(z),\, T_{j-1,j}[0] \big] \mathbb{B}(\bar{t})$$

$$= \mathbb{C}(\bar{s})\, T_{j-1,j}(z)\, \mathbb{B}(\bar{t})$$

$$= \mathcal{F}_{j-1,j}(z|\bar{s}; \bar{t}).$$

Symbolically, we will write: $\lim_{w\to\infty} \frac{w}{c} \mathcal{F}_{jj}(z|\bar{s}; \{w, \bar{t}\}) = \mathcal{F}_{j-1,j}(z|\bar{s}; \bar{t})$, $w \in \bar{t}^{(j)}$. Similarly, with the zero mode method, one gets the following relations:

$$\lim_{w\to\infty} \frac{w}{c} \mathcal{F}_{jj}(z|\bar{s}; \{w, \bar{t}\}) = \mathcal{F}_{j-1,j}(z|\bar{s}; \bar{t}), \qquad\qquad w \in \bar{t}^{(j)},$$

$$\lim_{w\to\infty} \frac{w}{c} \mathcal{F}_{jj}(z|\{w, \bar{s}\}; \bar{t}) = -\mathcal{F}_{j,j-1}(z|\bar{s}; \bar{t}), \qquad\qquad w \in \bar{s}^{(j)},$$

$$\lim_{w\to\infty} \frac{w}{c} \mathcal{F}_{j-1,j}(z|\bar{s}; \{w, \bar{t}\}) = \mathcal{F}_{j-2,j}(z|\bar{s}; \bar{t}), \qquad\qquad w \in \bar{t}^{(j-1)}, \tag{39}$$

$$\lim_{w\to\infty} \frac{w}{c} \mathcal{F}_{j,j-1}(z|\{w, \bar{s}\}; \bar{t}) = -\mathcal{F}_{j,j-2}(z|\bar{s}; \bar{t}), \qquad\qquad w \in \bar{s}^{(j-1)},$$

and so on. Thus, all the off-diagonal FF can be computed starting from diagonal ones. Moreover, from the limit

$$\lim_{w\to\infty} \frac{w}{c} \mathcal{F}_{j-1,j}(z|\{w, \bar{s}\}; \bar{t}) = \mathcal{F}_{j,j}(z|\bar{s}; \bar{t}) - \mathcal{F}_{j-1,j-1}(z|\bar{s}; \bar{t}), \quad w \in \bar{s}^{(j)} \tag{40}$$

one deduces that only one diagonal FF is needed to compute all the FF based on the monodromy matrix.

- *These considerations were developed in [43] for $Y(gl_3)$, but the same consideration can be done for $Y(gl_{m|p})$. Thus, the determinant representation for the scalar product (29) does generate determinant formulas for all FF in the models described by $Y(gl_3)$ [18, 44] and its super-analogs $Y(gl_{2|1})$ and $Y(gl_{1|2})$ [19]. However, a generalization of this method to the case of the $U_q(\widehat{gl}_n)$ algebra is not straightforward.*

### 5.3 Universal Form Factors

Consider the case of $Y(gl_n)$ or $Y(gl_{m|p})$ algebra. Let $\mathbb{C}(\bar{s})$ and $\mathbb{B}(\bar{t})$ be on-shell and such that their eigenvalues $\tau(z|\bar{s})$ and $\tau(z|\bar{t})$ are different. Then the ratio

$$\mathbb{F}_{i,j}(\bar{s}; \bar{t}) = \frac{\mathcal{F}_{i,j}(z|\bar{s}; \bar{t})}{\tau(z|\bar{s}) - \tau(z|\bar{t})} \tag{41}$$

is independent of $z$ and does not depend on the monodromy matrix vacuum eigenvalues. It depends solely on the $R$-matrix, and is thus model independent. We call it the universal FF.

One can show that the relations (39) yield similar relations for the universal FF. On the other hand, it follows form (35) that the diagonal universal FF are related to the twisted scalar product by

$$\mathbb{F}_{jj}(\bar{s}; \bar{t}) = \frac{d}{d\kappa_j} S^{\bar{\kappa}}(\bar{s}|\bar{t}) \Big|_{\bar{\kappa}=1}. \tag{42}$$

Thus, computing $S^{\bar{\kappa}}(\bar{s}|\bar{t})$ we can find all the universal FF.

Since the universal FF are completely determined by the *R*-matrix, they do not depend on the behavior of the monodromy matrix $T(z)$ at $z \to \infty$. Therefore, they can be used to calculate ordinary FF in models for which BAEs do not admit infinite roots. In this way, one can circumvent the $z \to \infty$ limit even for models where the zero modes method formally fails.

Note that in the models described by the $U_q(\widehat{gl}_n)$ algebra, the universal FF exist for the diagonal operators $T_{jj}(z)$ only.

## 5.4 Composite models

In the models, for which an explicit solution of the inverse scattering problem is known [45–47], the FF of the monodromy matrix entries immediately yield FF of local operators. In other cases, the FF of local operators can be calculated within the framework of the composite model [48]. In this model, the total monodromy matrix $T(z)$ is presented as a product of two partial monodromy matrices $T^{(2)}(z)$ and $T^{(1)}(z)$ as

$$T(z) = T^{(2)}(z)\, T^{(1)}(z) \tag{43}$$

with

$$\begin{aligned} T^{(2)}(z) &= \mathcal{L}_L(z) \cdots \mathcal{L}_{m+1}(z), \\ T^{(1)}(z) &= \mathcal{L}_m(z) \cdots \mathcal{L}_1(z), \end{aligned} \tag{44}$$

where $m \in [1, L[$ is an intermediate site of the chain. One can also consider continues composite models. Then the total monodromy matrix $T(z)$ is still given by (43), while the partial monodromy matrices $T^{(j)}(z)$ should be understood as continuous limits of the products of the $\mathcal{L}$-operators in (44).

We assume that each partial $T^{(j)}(z)$ possesses a pseudo-vacuum vector $|0\rangle^{(j)}$ so that $|0\rangle = |0\rangle^{(2)} \otimes |0\rangle^{(1)}$, and

$$T_{jj}^{(\ell)}(z)|0\rangle^{(\ell)} = \lambda_j^{(\ell)}(z)|0\rangle^{(\ell)}, \qquad \ell = 1, 2. \tag{45}$$

Similarly to how it was done in section 5.2, one can introduce partial zero modes $T_{ij}^{(\ell)}[0]$. Then in the models described by the Yangian $Y(gl_3)$, the FF of the first partial zero modes are related to the universal FF by

$$\mathbb{C}(\bar{s})T_{ij}^{(1)}[0]\mathbb{B}(\bar{t}) = \left( \prod_{k=1}^{2} \frac{r_k^{(1)}(\bar{s}^{(k)})}{r_k^{(1)}(\bar{t}^{(k)})} - 1 \right) \mathbb{F}_{i,j}(\bar{s};\bar{t}), \tag{46}$$

where

$$r_k^{(1)}(u) = \frac{\lambda_k^{(1)}(u)}{\lambda_{k+1}^{(1)}(u)}, \tag{47}$$

and we used the shorthand notation (8) for the products of these functions. It is assumed in (46) that the on-shell BVs $\mathbb{C}(\bar{s})$ and $\mathbb{B}(\bar{t})$ have different eigenvalues.

Since the number $m$ of the intermediate site is not fixed, the FF of the first partial zero modes give an immediate access to the FF of the local operators $(\mathcal{L}_m)_{ij}[0]$ due to

$$T_{ij}^{(1;m)}[0] = \sum_{k=1}^{m} (\mathcal{L}_k)_{ij}[0], \tag{48}$$

where we have stressed by the additional superscript $m$ that the partial zero mode $T_{ij}^{(1;m)}[0]$ depends on $m$. Then

$$\mathbb{C}(\bar{s})(\mathcal{L}_m)_{ij}[0]\mathbb{B}(\bar{t}) = \mathbb{C}(\bar{s})\Big( T_{ij}^{(1;m)}[0] - T_{ij}^{(1;m-1)}[0] \Big)\mathbb{B}(\bar{t}). \tag{49}$$

- *These calculations for the Yangian $Y(gl_3)$ can be found in [49–52], with application to the two-component Bose gas. Similar equations for FF of local operators in the case of Yangians $Y(gl_{2|1})$ and $Y(gl_{1|2})$ was obtained in [53]. Most probably, FF of local operators in the general $Y(gl_n)$ and $Y(gl_{m|p})$ cases can be expressed in terms of the universal FF in the same way.*

# 6 Conclusion

Concerning the points described in the present review, many directions remain to be developed. Among them, one can distinguish the following ones.

(i) Finding a simpler expression for the scalar product of off-shell BVs. We have already mentioned the determinant expressions that seem to be well-adapted for the calculation of correlation functions and for the thermodynamic limit. Such expressions, even in the case of $U_q(\widehat{gl_3})$, are thus highly desirable. On this point, note the determinant expression for XXX model in the thermodynamic limit found by Bettelheim and Kostov [54]. Remark also the approach by N. Grommov *et al.* using a single 'B'-operator [55], for $Y(gl_n)$ with fundamental representations.

(ii) Another way to get simple expressions for scalar products could be to use an integral representation. A first step has been done by M. Wheeler in [17]. Remark also that the projection method in the current presentation provides an integral representation, see e.g. [56].

(iii) Once determinant expressions are known for scalar products, in the case of (super) Yangians, the zero mode methods allows to get similar expressions for the form factors. It would be good to get a similar method of the $U_q(\widehat{gl_n})$ case. It seems that the zero mode methods can be adapted to this case: we hope to come back on this point in a further publication.

Obviously, all these points are the first step towards the complete calculation of correlation functions and their asymptotics. As mentioned in the introduction, this calculation depends specifically on the model one wishes to study. Among the possible applications, one can distinguished multi-component Bose gas, tJ-model or the integrable approach to amplitudes in Super-Yang-Mills theories.

Finally, the case of other quantum algebras, based on orthogonal or symplectic Lie algebras is also a direction that deserved to be studied.

# Acknowledgments

We would like to thank the organizers of the JM Maillet's 60s birthday workshop for given us the opportunity to present this review. We take also this opportunity to wish again to Jean-Michel a happy birthday.

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
