# Peer review of "Nested Algebraic Bethe Ansatz in integrable models: recent results"

_SciPost Physics Lecture Notes, doi:SciPost Phys. Lect. Notes 6 (2018)_

## Round 1 · Referee Report · Anonymous (Referee 1) · 2018-5-21

Strengths

1- Clear and compact review of the known the determinant formulas for overlaps of Bethe vectors and form-factors in solvable models with higher rank symmetries.

2- Useful for readers interested mainly in the application of these results.

Weaknesses

1- No weaknesses

Report

This paper gives a useful concise review of the exact results obtained by the authors in the recent years concerning the Bethe vectors and their overlaps in integrable models associated with higher rank algebras, including super algebras. This is a valuable paper and I recommend publication in its present form.

Requested changes

no requested changes

---

## Round 1 · Referee Report · Anonymous (Referee 2) · 2018-6-30

Strengths

The manuscript is the review about the netted algebraic Bethe ansatz based on already published papers of the authors and the others.

Weaknesses

Although the paper focuses on the outline of the nested algebraic Bethe ansatz without technical details, it is worth publishing as proceedings.

Report

The authors have been working on this topic for a long time and the manuscript is well-summarized. All the technical details are removed and it is hard to follow the equations, but this would help for people to understand what the nested algebraic Bethe ansatz is. There are several points I want to remark.

Requested changes

In the section 2, “The transfer matrix t(z)” - In the equation (6), the authors use the notation (-1)^{[i]} which is not defined. This is the common notation in the field of superalgebra, but it will be helpful for non-specialists if the definition is given.

In the section 3, “In the case of higher rank n” - Since the manuscript is mostly devoted to reviews, it is better to give references for the equations (11)-(13).

In the section 3, “Known formulas: the trace formula” - Probably this is rather clear, but it is helpful if the authors denote the space where the elementary matrices e_{i,j} acts. - Give the reference about a Gauss decomposition of the monodromy matrix.

In the section 4, “Sum formula” - Although the authors say the coefficient W_{part} are rational functions of the Bethe parameters, this is true only for the non q-deformed cases. Since the q-deformed cases are also mentioned in the same subsection, some comments about the trigonometric cases are seemed to be required. - Also, give the references for the formula (24). (They are given in the comments later, but it is better to give them at the beginning when the formula first shows up.)

In the section 4, “Determinant formula” - The determinant formula is given under “restrict conditions” as is mentioned in the manuscript. What this condition means the special twist for the transfer matrix, but this becomes clear only after reading the manuscript at the end of this subsection. It is better to remark what is the condition the authors consider here.

In the section 5, “Bethe vectors and zero modes” - The reason why the authors used the brackets in the equation (38) is to emphasize that one of the Bethe parameters of the family (j-1) is sent to infinity. If so, the limit in the next sentence should be t_k^{(j-1)} \to \infty instead of t_k^{(j)} \to \infty,

In the section 5, “Bethe vectors and zero modes” - In the footnote, several problems are remarked and the authors say “we show in section 5.3 how this problem can be solved”. However, what is explained is why the method fails and I found so solution to this problem. Or, do the authors mean that the formula for the universal form factors is the solution? (As they do not depend on the behavior of the spectral parameters. )

---

## Round 2 · List of Changes

We thank the referee for his comments. We have modified our manuscript accordingly. More precisely, and considering the different questions raised by the referee:

$\bullet \textit{In the section 2, The transfer matrix t(z)}$ We added a sentence right after eq. (6)

$\bullet \textit{In the section 3, In the case of higher rank n}$ We added references just before eq (12).

$\bullet \textit{In the section 3, Known formulas: the trace formula}$ - We mentioned that the matrices $e_{ij}$ act in $\mathbb{C}^3$. - We added a reference in the sentence mentioning the Gauss decomposition

$\bullet \textit{In the section 4, Sum formula}$ - We disagree with the referee about the quantum case: the coefficients $W_{part}$ are still rational functions of the Bethe parameters. We guess the referee had in mind a presentation with additive spectral parameters (for which indeed the coefficients $W_{part}$ are not rational functions anymore), but we use multiplicative ones, for which we still have rational functions. To clarify this point we gave the form of the functions $f$ and $g$ in the quantum case, see the new equation at the beginning of section 2.1. - The common guideline of the article is to have a special paragraph were the references are gathered, so we prefer to keep it this way. However, we mention after eq (24) the first peoples that got such a formula, and refer to the special paragraph.

$\bullet \textit{In the section 4, Determinant formula}$ In fact the whole paragraph is needed to specify what are the special conditions (we need to introduce the twisted transfer matrix, the twisted BVs, etc..). Then, we don't see how to fulfill the requirement of the referee. Instead, we added sentences at the end of the first paragraph to mention that the special requirements will be shown at the end of the subsection.

$\bullet \textit{In the section 5, Bethe vectors and zero modes}$ - We corrected $j$ in $j-1$ in the limit $t^{(j-1)}_k\to\infty$. - Indeed, the universal FF are the solution, as the referee guessed. We thought it was clear from our section 5.3, but apparently it was not. We modified the end of this section to be clearer.

---

## Editorial Decision

published